# Does Environmental Regulation Have an Employment Dividend? Evidence from China

**Chao Wu** [1,2,*] **and Yu Hua** [1]

1    School of Finance and Economics, Jiangsu University, Zhenjiang 212013, China
2    School of Economics, Fudan University, Shanghai 200433, China
*    Correspondence: chao_wu@fudan.edu.cn

**Abstract:** Environmental regulations not only benefit environmental improvement but may also have a potential employment dividend, which is important for sustainable economic development. Based on the provincial panel data of China from 1997 to 2019, the spatial Dubin model is used to analyze the spatial spillover effects of environmental regulation on employment. From the findings, there is a significant spatial agglomeration characteristic in the employment scale. The environmental regulation positively influences the local employment scale, while there is a negative spatial effect on employment in neighboring regions. With regards to the regional heterogeneity analysis, environmental regulation has a spatial negative correlation with employment in coastal areas and less polluted areas. Additionally, in inland areas and less polluted areas, environmental regulation has a significant positive impact on the employment scale. Meanwhile, market-based environmental regulation has the potential to expand the employment scale in local and neighboring areas, while command-and-control environmental regulation impacts negatively on employment in neighboring areas. This study found that environmental regulation has an employment dividend. These findings reveal the spatial dependence between environmental regulation and employment, which will help policy makers consider the environmental and employment effects of environmental regulation more comprehensively. Therefore, the government should formulate targeted policies under regional differences to promote high-quality employment and construct a multiple governance environmental regulatory system.

**Keywords:** environmental regulation; employment; spatial Durbin model; heterogeneity analysis

## 1. Introduction

In the 21st century, human-induced climate change is one of the major challenges for all economies. To confront the increasing threat of global warming, environmental regulation (ER) emerged as an important policy tool [1]. As a responsible country, China has introduced a number of ER aimed at developing an environmentally friendly, resource-saving and recycling society. In order to accelerate the improvement of environmental quality, the Chinese government launched the "three-year Action Plan to fight air pollution". After three years of efforts, the emission of primary air pollutants has been greatly reduced, and the air quality has been improved [2]. However, new environmental regulations significantly reduced manufacturing labor demand by about 3 percent [3]. According to the Ministry of Environmental Protection, as of June 2017, more than 100,000 enterprises in the Beijing–Tianjin–Hebei regions faced closure or suspension of production due to pollution emissions. This may lead to a large number of unemployment and cause social problems.

In 2015, to harmonize employment and the environment, United Nations included these 2 in the 17 Sustainable Development Goals. The eighth goal of the SDGs is set to encourage persistent, comprehensive and sustainable economic progress, decent work and full and productive employment for all. Goal 12 is focused on countries taking deliberate steps to improve education, environmental awareness of the people, human capital and

institutional capacity and to implement climate mitigation, early warning and impact reduction technologies to achieve sustainable economic growth without damaging the environment. Therefore, as the world's most populous country, to pursue its sustainable economic growth and the targets of 2030 UN agenda, the impact of ER on employment is an issue that needs urgent discussion.

On this subject, previous literature provides two distinctive streams of academic debate which are "innovation compensation theory" and "compliance cost theory". The former provides a positive relationship between the two variables, implying that environmental regulation will make firms adopt their external cost and expedite innovation of firms [4]. Chen et al. [5] believe that environmental regulation can influence labor demand for employment by augmenting technological progress and innovation. On the contrary, the latter claims that environmental cost hinders the productivity efficiency and international competitiveness of firms, and, thus, the employment rate declines [6]. There is a plethora of literature that shows that stringent environmental regulation negatively impacts labor's wage and employment rate as a result of low production and increased compliance cost [7]. Based on the premise, the first objective of this study is to explore the impact of ER on employment growth in China.

A well-crafted environmental regulation may trigger the labor flow among different regions and different industries. Due to the "GDP competition", environmental regulation may also exert a "race to bottom" effect [8,9]. Simply put, governments are inclined to be lenient with ER so as to attract investors and promote economic growth. Now, this may result in industrial transfer as well as labor mobility among different regions. Sun et al. [10] found that ER caused the transfer of labor from big cities to small ones and increased the employment rate of the primary and tertiary industries in small cities. On this premise, the second objective of this study is to examine the spatial spillover effect of ER on employment in neighboring provinces. It has a far-reaching policy significance.

China's ER can be divided into command-and-control environmental regulation (CER), market-based environmental regulation (MER) and voluntary environmental regulation (VER) [11]. Yu and Zhang [12] found that strict CER were associated with a sharp decline in labor demand. Firms in cities with more stringent ER have experienced greater labor reductions compared to firms in cities with less stringent ER [13]. As opposed to CER, MER is generally thought to expand the employment scale [14,15]. VER is less discussed in relation to employment. The third objective of this study is to explore the different spatial effects of heterogeneous ER on employment.

Specifically, this paper uses China's provincial panel data from 1997 to 2019 and adopts traditional OLS model, two-way fixed effect model, spatial autoregressive model (SAR), spatial error model (SEM) and spatial Durbin model (SDM) to explore the spatial effects of ER on employment. It is found that, first of all, ER is conducive to the increasing the employment scale but has a negative impact on the employment scale of neighboring provinces. Secondly, ER has a negative spatial effect on employment in coastal areas and less polluted areas but has a significant positive effect on employment in inland areas and less polluted areas. Thirdly, from the perspective of heterogeneous ER, it is found that MER has a significant promoting effect on the employment scale in local and surrounding areas, while CER only has a significant negative effect on employment in surrounding provinces.

Though extensive studies have focused on this subject, their results are inconclusive. One of the potential reasons is that the existing literature might have ignored the spatial dependence of both the dependent variable and its explanatory indicators. The following potential academic contributions are going to be made by this study. Firstly, to avoid the estimation biases through ignoring the spatial effects, we employ SDM to analyze the environmental regulation–employment nexus in China. This method would likely display a clear picture of the spatial relationship between the two variables. Moreover, considering the significant heterogeneity among different regions, this paper prolongs the existing body of research by studying the impact of ER in different locations and pollution levels on employment, which might help the government implement the targeted policies. Last,

this paper breaks down ER into three categories and examines their individual impact on employment. This approach enriches the existing studies on ERs and labor demand and reveals the internal mechanism by which ER affect employment.

This study is consequently organized as follows. Relevant literature is reviewed in Section 2. Section 3 is methodology and data. Results and discussion are explained in Section 4. Section 5 debates conclusions and policy implications.

## 2. Literature Review and Theoretical Analysis

### 2.1. The Compliance Cost Effect

Theoretically, the compliance cost hypothesis holds that firms will raise prices in order to uphold the profit due to increased costs associated with ER, which leads to lower market demand and a reduction in production scale, resulting in higher unemployment (this is called output effect) [16,17]. More specifically, ER inevitably influences the operational cost of manufacturing firms and decreases the tangible incomes of relevant firms [18]. Zhang et al. [19] studied pollution-intensive industries (PIIs) and found that polluters are more willing to reduce or stop production in response to CER, and the increase in compliance cost is the main intermediary factor.

The substitution effect of compliance costs has a positive or negative impact on employment [20]. The costly equipment upswings both operation and production costs, which forces companies to realign their factors input. The implementation of developed technologies can augment firms' efficiency, substituting labor consequently and decreasing employment [21]. Chen et al. [22] found that the application of the $SO_2$ emission trading system impedes a firm's labor investment efficiency, and the impact is driven by the firm's over-firing or under-hiring behaviors. Sheng et al. [23] claimed that robust environmental regulation has decreased employment in the manufacturing firms by both substitution and output effects across 18 cities in China. Zheng et al. [6] point out that ER would significantly reduce the labor demand of enterprises, which comes from the reduction of production scale and the substitution of human capital. On the other hand, in order to comply with new and stricter ER, firms must hire workers to install and maintain equipment and participate in environmental management activities, which may employ more workers than before the regulation [24]. By distinguishing between high-skilled and low-skilled labor, Zhong et al. [25] found that the compliance cost effect of ER would promote the employment of high-skilled labor while inhibiting the employment of low-skilled labor.

### 2.2. The Spatial Effect

The spillover effect of ER on employment mainly comes from the following two aspects. On the one hand, gradient differences in ER will lead to industrial transfer [26], thus increasing the employment scale in the industrial receiving regions [10]. Theoretically, ER affects the location choice of PIIs through cost and innovation effects [27]. Driven by compliance costs, manufacturers are more willing to relocate to inland areas with looser ER [28]. This can save environmental costs and reduce the cost of negotiations with governments, NGOs and residents. Industrial transferees restructure local industries to improve the environment, while industry recipients achieve employment growth within the environmental capacity [29].

On the other hand, due to the strategic interaction between local governments, ER may indirectly affect the employment situation in the surrounding areas. In other words, ER may have spatial spillover effect on the labor market in the surrounding areas. Firstly, local ER may affect the formulation of ER in neighboring areas. Under the assessment mechanism of environmental tournaments, local governments compete to improve environmental standards in order to obtain better rankings, forming a "Race to Top" mode. Tightening ER will force local firms to use cleaner technologies, increasing demand for labor. Secondly, the Chinese government has launched the "Joint Prevention and Control System of Regional Air Pollution" to coordinate the environmental quality of the region [30]. The regions with

high ER intensity will produce more learning effects on the surrounding regions and thus have a synergistic effect on the labor market of the surrounding provinces [31].

*2.3. The Innovation Compensation Effect*

The innovation compensation effect claims that ER can influence labor demand for employment by augmenting technological progress and innovation [5]. Therefore, ER may not be the reason for unemployment. According to this hypothesis, well-crafted ER facilitate innovation that may cover the costs of meeting regulatory requirements, promote competitiveness and generate job opportunities [4,32]. Ren et al. [33] found that an emission trading program helps drive firms to expand the production scale, significantly increasing the labor demand of regulated firms. Numerous researchers have analyzed the environmental regulation–innovation interplay from the perspective of the innovation compensation effect. Luo et al.'s [11] study validated innovation compensation effect in China through exploring the nexus between ER and green innovation. Regarding the positive effect of technology innovation on employment, numerous scholars have carried out extensive research. For example, Li et al. [1] found that technological innovation improves the enterprise employment. Zhu et al. [34] found that process innovation significantly increases the employment.

Furthermore, Acemoglu (2003) [35] put forward the notion of biased technological change theory, including labor-biased technical change and capital-biased technical change. This concept has drawn greater attention of scholars, as it can classify various crucial problems, including labor employment structure, change in environmental technology and income gap among countries [36]. Song et al. [37] argued that ER can promote the progress of environmental-biased technology, and the positive effect of enterprise competitiveness will be stronger than the negative effect caused by the reduction of production scale, thus increasing the labor demand of enterprises.

In summary, previous studies have considered the impact of ER on employment from the perspective of compliance costs and innovation compensation effect, which laid the theoretical foundation for this study. However, there is still room for further research. First, previous studies have only analyzed individual aspects of compliance costs or innovation compensation effects and have not integrated the two into a unified research framework. Secondly, less attention has been paid to this area of research from the spatial spillover effect. Finally, the compliance cost effect and innovation compensation effect of heterogeneous ER are different, which lacks analysis from the perspective of ER heterogeneity. Hence, the potential academic contribution of this research is to explore their relationships from spatial and heterogeneity perspectives. It may provide a new explanation for the impact of environmental regulation on employment, which is of great significance. The theoretical reasonings for the impact of ER on employment are shown in Figure 1.

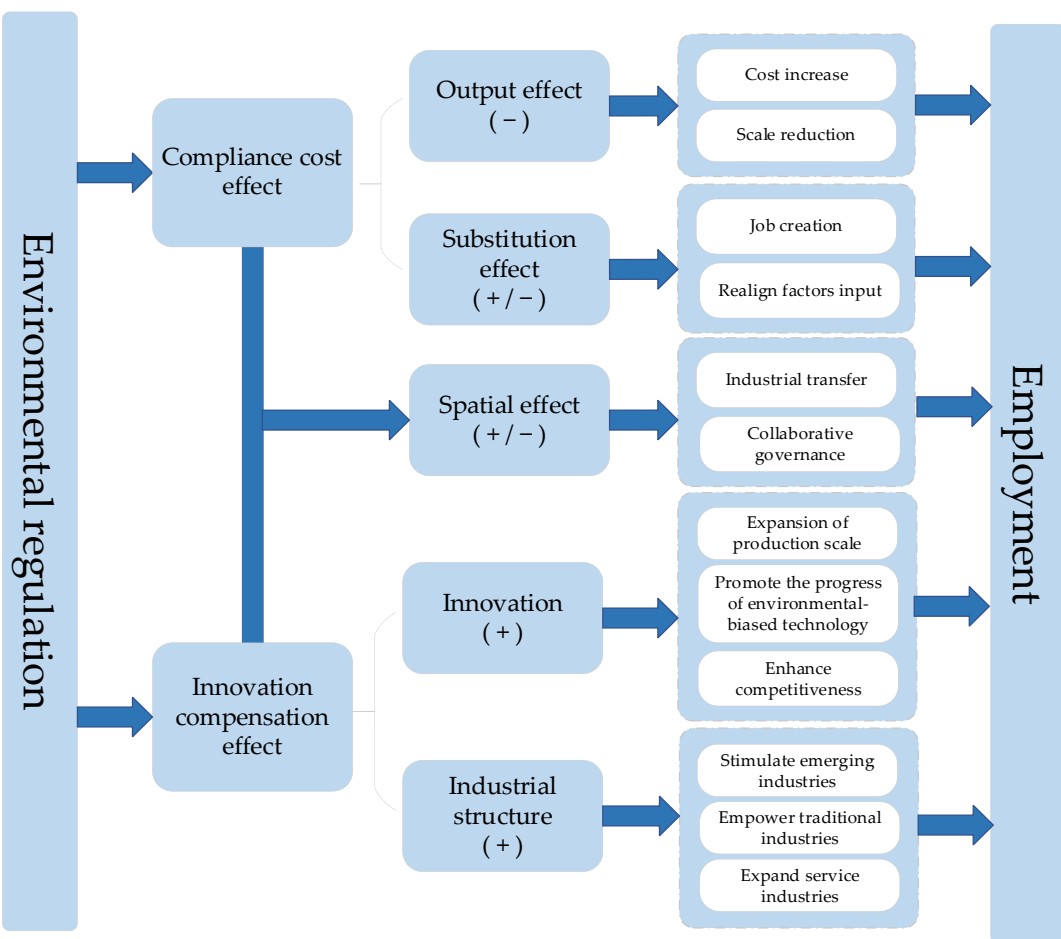

**Figure 1.** Mechanism analysis diagram.

## 3. Methodology and Data

### 3.1. Spatial Autocorrelation Test

To check the spatial characteristics and autocorrelation of the employment and environmental regulation, global Moran's *I* and Geary's *C* indexes are employed in this study [38] (see Equations (1) and (2)).

$$I = \frac{n \sum_{i=1}^{n} \sum_{j=1}^{n} W_{ij}(x_i - \overline{x})(x_j - \overline{x})}{\sum_{i=1}^{n} \sum_{j=1}^{n} W_{ij}(x_i - \overline{x})^2} = \frac{\sum_{i=1}^{n} \sum_{j=1}^{n} W_{ij}(x_i - \overline{x})(x_j - \overline{x})}{s^2 \sum_{i=1}^{n} \sum_{j=1}^{n} W_{ij}} \quad (1)$$

$$C = \frac{(n-1)\sum_{i=1}^{n} \sum_{j=1}^{n} w_{ij}(x_i - x_j)^2}{2(\sum_{i=1}^{n} \sum_{j=1}^{n} w_{ij})\left[\sum_{i=1}^{n}(x_i - \overline{x})^2\right]} \quad (2)$$

whereas $x_i$ is the variable in $i$th province, $\overline{x}$ is the mean value of $x$, $n$ is the amount of the provinces in China while $W_{ij}$ denotes the spatial weight matrix. When Geary's *C* indexes are closer to 0 or 2, and Moran's *I* index is closer to 1 or −1, a strong spatial autocorrelation between the variables is suggested.

Regarding the spatial weight matrix, this research adopts a geographic distance matrix to quantify the locational factors between provinces, and the geographic distance matrix is set as follows.

$$W_d = \begin{cases} 1/d_{ij}, & i \neq j \\ 0, & i = j \end{cases} \quad (3)$$

whereas $W_d$ is the geographic distance matrix, and $d_{ij}$ denotes the distance between $i$th and $j$th province. The greater distance between provinces leads to the smaller interregional influence.

Moreover, the local Moran's *I* index is also utilized to analyze the spatial characteristics of different provinces. The formula is expressed in Equation (4).

$$I_i = \frac{(x_i - \overline{x})}{s^2} \sum_{j=1}^{n} W_{ij}(x_j - \overline{x}) = z_i \sum_{j=1}^{n} W_{ij} z_j \tag{4}$$

where $z_i$ and $z_j$ are the deviations between the values of *i*th, *j*th province and mean value, respectively.

### 3.2. Spatial Econometric Models

This paper mainly analyzes the spatial effects of ER on the labor market in China. The basic econometric model is constructed in Equation (5).

$$\ln Emp_{it} = \beta_0 + \beta_1 ER_{it} + \alpha_i \ln X_{it} + \delta_i + \varepsilon_{it} \tag{5}$$

where $Emp_{it}$ represents employment scale, $ER_{it}$ denotes environmental regulation, and $X_{it}$ indicates control variables.

However, the results based on traditional OLS estimations might be biased if the spatial effects of the variables are ignored. Therefore, this study uses the SDM to investigate the spatial spillover effect of ER on employment in China. The SDM model can measure the spillover effects of dependent variable as well as explanatory variables [39]. The expression of SDM is:

$$y = \rho W y + X\beta + WX\theta + \varepsilon \tag{6}$$

where $\rho$ is the spatial autoregressive coefficient, and $\beta$ and $\theta$ are the direct coefficient and the spatial coefficient of the independent variable, respectively.

This paper constructs the geographical distance matrix and sets the SDM model of ER on employment according to the benchmark model as follows:

$$\ln Emp_{it} = \rho W \ln Emp_{it} + \beta_1 ER_{it} + \alpha_i \ln X_{it} + \gamma_1 \times W \ln ER_{it} + \theta_i \times W \ln X_{it} + \delta_i + \varepsilon_{it}$$
$$\varepsilon_{it} \sim N(0, \sigma_{it}^2 I_n) \tag{7}$$

However, the SDM model might also lead to an endogenous problem because it captures the effects of spatial lag dependent and independent variables simultaneously [40]. Hence, following the study of Pace et al. [41], we employ maximum likelihood estimation to address this problem.

### 3.3. Data

A panel provincial data of China from 1997 to 2019 is used in this study. All data related to price index are deflated at the 1997 price. The original data of each province were obtained from the China Statistical Yearbook of Environment, China Statistical Yearbook.

The dependent variable is employment scale (*Emp*), which is proxied by the number of employees in urban areas. ER is the core explanatory variable. Compared with using a single indicator to reflect the intensity of ER, the indicator system can measure the ER more comprehensively. According to Luo et al. [11], this study segmented ER into three dimensions, CER, MER and VER, and selected corresponding indicators to construct an indicator system (see Table A1). The entropy TOPSIS method is utilized as the evaluate technique in this research.

Moreover, human capital (*HC*), foreign direct investment (*FDI*), patent application (*PAT*) and gross domestic product (*GDP*) are introduced as control variables to confront the issue of variable bias.

According to Ma et al. [42], *HC* is measured by Equation (8).

$$HC_{it} = 6PS_{it} + 9JS_{it} + 12SS_{it} + 15CE_{it} + 16UE_{it} + 19ME_{it} \tag{8}$$

where $PS_{it}$, $JS_{it}$, $SS_{it}$, $UE_{it}$, $CE_{it}$ and $ME_{it}$, are the proportion of the employees with primary, junior, high school, college, university education and master education, respectively. The

numerical values 6, 9, 12, 15, 16 and 19 denote years of education. Increase in human capital is positively related to productivity and business success, which might produce more job opportunities [43]. However, enriched human capital might also crowd out the low-quality employment, which will decrease the employment scale and ultimately widen China's wealth gap. Some scholars based on China data held the view that human capital played either no role on employment or negatively affected employment [44].

The FDI is selected to test the impacts of China's opening-up policy on employment. FDI might have two-way effects on employment. On one hand, Rong et al. [45] found that 1% increase in FDI can increase 0.216% of employment expectation. However, FDI had a certain degree of crowding-out effect on the employment of host countries [46].

Following Kim et al. [47], the number of patents applied for (PAT) is used to proxy the technology level; with the development of advanced technologies, more job opportunities are created and the employment scale is expanded. The research of Van Roy et al. [48] proved that technological innovation creates jobs. However, this effect is only significant for high-tech enterprises. In contrast, advanced technologies can develop high-efficiency machine equipment, which exerts a negative effect on employment. Buerger et al. [49] showed that technical progress crowds-out employment in the German chemical industry sector from 1999 to 2005.

GDP in this paper has been used to probe the influence of economic growth on labor. GDP is positively correlated to employment. Ghosh [50] established that GDP growth was one of the reasons why India's employment level rose higher. However, some scholars believe that there is decoupling between GDP and employment growth in recent years [51]. Table 1 presents the descriptive analysis.

**Table 1.** Descriptive statistics.

| Variable | Definition | Obs. | Unit | Mean | Std. Dev. | Min | Max |
|---|---|---|---|---|---|---|---|
| Emp | Employment scale | 690 | 1000 persons | 4668.49 | 3125.21 | 425 | 20,646 |
| ER | Environmental regulation | 690 | - | 0.213 | 0.195 | 0 | 1 |
| HC | Human capital | 690 | % | 886.220 | 129.729 | 471 | 1390.1 |
| FDI | Opening-up level | 690 | $10^6$ CNY | 4219.949 | 5500.811 | 4.46 | 29,039.96 |
| PAT | Technology level | 690 | item | 44,462.61 | 93,540.05 | 124 | 807,700 |
| GDP | Economic level | 690 | $10^9$ CNY | 13,596.62 | 16,061.23 | 202.050 | 107,671.1 |

*3.4. Spatial Distribution of Employment Scale and Environmental Regulation*

Figure 2a,b show the geographical distribution of the employment scale in China in 1997 and 2019, respectively. It can be seen that the provinces with large employment scale in China are mainly eastern provinces, such as Shandong, Jiangsu and Guangdong. It is mainly due to the superior geographical position and advanced economic development. Compared with 1997, it is obvious that the employment scale in north-eastern regions (Liaoning, Heilongjiang and Jilin) went down in 2019, and employment areas were more concentrated.

Figure 3a,b show the geographical distribution of ER intensity in 1997 and 2019, respectively. It obvious that ER has a significant spatial agglomeration characteristic. Overall, the intensity of ER in China is low in the west and high in the east, which may be related to China's opening-up strategy and ecological civilization construction policy.

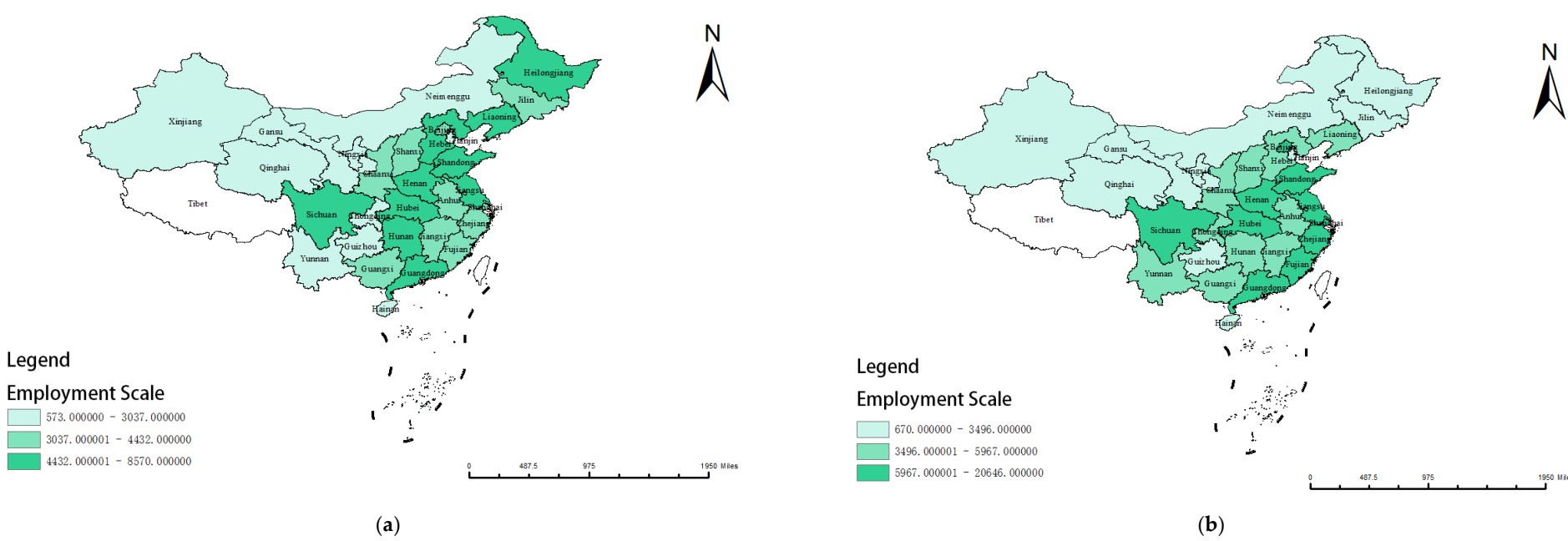

**Figure 2.** (**a**) Employment scale_1997; (**b**) Employment scale_2019.

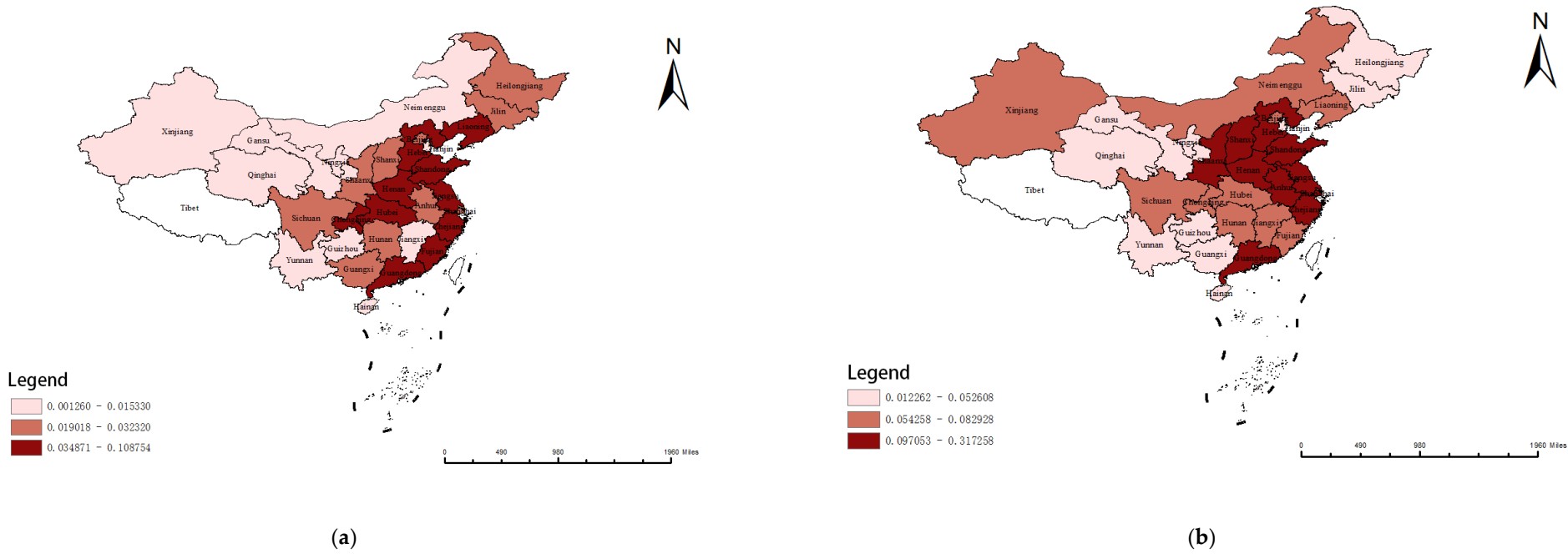

**Figure 3.** (**a**) Environment regulation_1997; (**b**) Environment regulation_2019.

## 4. Discussion

### 4.1. Spatial Correlation Analysis

Based on the geographic distance matrix, Moran's I and Geary's c methods are adopted to examine the spatial correlation of employment scale in different regions of China. It can be found that the global Moran's I and Geary's c index of employment scale is significantly positive at 5% significance level from 1997 to 2019 in Table 2, which indicated that the employment scale had significant spatial correlation.

**Table 2.** Spatial correlation analysis of employment scale.

| Year | lnEmp | | | | Year | lnEmp | | | |
| --- | --- | --- | --- | --- | --- | --- | --- | --- | --- |
| | Moran's I | p | Geary's C | p | | Moran's I | p | Geary's C | p |
| 1997 | 0.029 ** | 0.023 | 0.891 ** | 0.012 | 2009 | 0.026 ** | 0.030 | 0.902 ** | 0.019 |
| 1998 | 0.029 ** | 0.023 | 0.891 ** | 0.012 | 2010 | 0.027 ** | 0.028 | 0.902 ** | 0.019 |
| 1999 | 0.029 ** | 0.023 | 0.891 ** | 0.012 | 2011 | 0.040 ** | 0.011 | 0.889 ** | 0.010 |
| 2000 | 0.028 ** | 0.025 | 0.897 ** | 0.017 | 2012 | 0.043 *** | 0.008 | 0.883 ** | 0.007 |
| 2001 | 0.027 ** | 0.028 | 0.892 ** | 0.014 | 2013 | 0.035 ** | 0.015 | 0.893 *** | 0.012 |
| 2002 | 0.024 ** | 0.033 | 0.894 ** | 0.016 | 2014 | 0.038 ** | 0.012 | 0.891 ** | 0.011 |
| 2003 | 0.025 ** | 0.032 | 0.897 ** | 0.017 | 2015 | 0.038 *** | 0.012 | 0.891 ** | 0.011 |
| 2004 | 0.024 ** | 0.033 | 0.893 ** | 0.015 | 2016 | 0.037 ** | 0.013 | 0.893 ** | 0.012 |
| 2005 | 0.023 ** | 0.037 | 0.897 ** | 0.018 | 2017 | 0.034 ** | 0.017 | 0.898 ** | 0.015 |
| 2006 | 0.025 ** | 0.032 | 0.897 ** | 0.017 | 2018 | 0.039 ** | 0.011 | 0.891 ** | 0.10 |
| 2007 | 0.025 ** | 0.033 | 0.898 ** | 0.017 | 2019 | 0.041 *** | 0.009 | 0.888 *** | 0.009 |
| 2008 | 0.026 ** | 0.030 | 0.899 ** | 0.018 | | | | | |

Notes: *** $p < 0.01$, ** $p < 0.05$.

In Figure 4, the scatterplot of local Moran's I index is divided into four quadrants. The spatial dependence of provinces in the first and the third quadrants is positive, while the units in second and fourth quadrants are negatively spatially correlated. The circles in the figure represent the spatial dispersion of the provinces with the employment scale, and the coefficients of the scatter fitting lines are Moran's I index. It can be found that the clusters of provinces regarding employment scale in 1997 and 2019 are mainly located in the first and third quadrants, indicating that the employment scale had significant local spatial agglomeration characteristics. In summary, it is appropriate to use the spatial econometric model in this study and not result in a bias in the regression results.

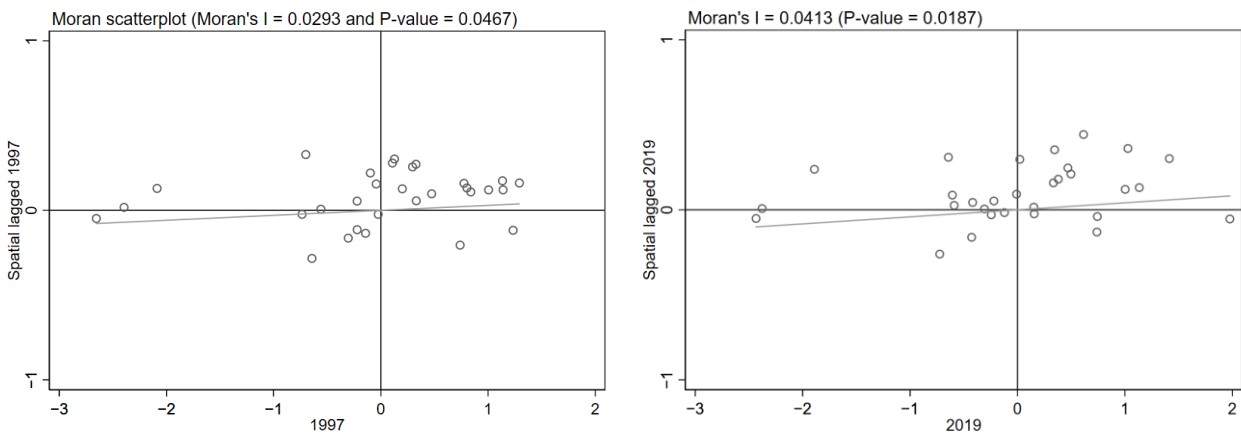

**Figure 4.** Local Moran's I scatterplot of employment scale in 1997 and 2019.

### 4.2. Spatial Effect Estimation Results

The spatial spillover effects of ER on employment scale are analyzed in Table 3. The results of both LM (Lagrange Multiplier) and robust LM tests are positive and pass the significance test of 5%. The null hypothesis of the panel model with no-spatial effect is

rejected, which indicates that the residual estimated by the model has spatial autocorrelation. In addition, the Wald and LR tests were further utilized to justify the suitable spatial econometric model, and the results of these tests suggest that the SDM model is much more appropriate compared to SAR and SEM, respectively. The Hausman tests of the three models rejected the null hypothesis. Hence, the SDM model with fixed effects is selected. In order to compare, this paper also provides the traditional OLS, fixed effects model, SAR and SEM estimation results of the model.

**Table 3.** The estimation results of the benchmark regression.

| | POLS | FE | SAR | SEM | SDM |
|---|---|---|---|---|---|
| ER | 0.543 *** | 0.177 *** | 0.188 *** | 0.191 *** | 0.169 *** |
| | (0.1295) | (0.0594) | (0.0551) | (0.0552) | (0.0527) |
| lnHC | −1.102 *** | 0.362 *** | 0.330 *** | 0.314 ** | 0.124 |
| | (0.1538) | (0.1318) | (0.1224) | (0.1248) | (0.1187) |
| lnPAT | 0.0777 ** | 0.118 *** | 0.103 *** | 0.0958 *** | 0.0640 *** |
| | (0.0322) | (0.0159) | (0.0149) | (0.0155) | (0.0147) |
| lnGDP | 0.293 *** | 0.170 *** | 0.177 *** | 0.199 *** | 0.222 *** |
| | (0.0488) | (0.0455) | (0.0422) | (0.0439) | (0.0417) |
| lnFDI | 0.0721 *** | −0.0269 *** | −0.0293 *** | −0.0297 *** | −0.0355 *** |
| | (0.0165) | (0.0091) | (0.0085) | (0.0084) | (0.0082) |
| $\rho$ | | | 0.560 *** | | 0.339 *** |
| | | | (0.0803) | | (0.1093) |
| W×ER | | | | | −0.858 ** |
| | | | | | (0.3690) |
| W×lnHC | | | | | 1.134 |
| | | | | | (0.8171) |
| W×lnPAT | | | | | 0.815 *** |
| | | | | | (0.0943) |
| W×lnGDP | | | | | −0.988 *** |
| | | | | | (0.2187) |
| W×lnFDI | | | | | 0.223 *** |
| | | | | | (0.0784) |
| $R^2$ | 0.6467 | 0.9754 | 0.8580 | 0.8994 | 0.4286 |
| Log-likelihood | | | 519.9470 | 515.0474 | 571.3207 |
| LM test no spatial lag | | | | | 25.075 *** |
| Robust LM test no spatial lag | | | | | 487.016 *** |
| LM test no spatial error | | | | | 733.876 *** |
| Robust LM test no spatial error | | | | | 1195.817 *** |
| Wald_spatial_lag | | | | | 12.62 ** |
| LR_spatial_lag | | | | | 104.49 *** |
| Wald_spatial_error | | | | | 13.13 ** |
| LR_spatial_error | | | | | 104.49 *** |
| Hausman | | | | | 109.36 *** |
| FE | NO | YES | YES | YES | YES |
| Observations | 690 | 690 | 690 | 690 | 690 |

Notes: *** $p < 0.01$, ** $p < 0.05$. The Standard errors are in parentheses.

In Table 3, the coefficients of ER in SDM model are significantly positive and consistent with the estimated results of POLS, FE, SAR and SEM, suggesting that the intensifying local environmental regulation will increase the employment scale. The stricter local environmental regulation is likely going to create more jobs for environmental protection. According to the innovation compensation effect, strict ER is conducive to forcing pollution-intensive enterprises to transform and upgrade through technological progress and innovation, thus creating more job opportunities. Several studies have reached the same conclusion as this paper [33,37,52]. $\rho$ is statistically positive, which indicates that there is a spatial positive spillover effect of the employment scale in China. This is reflected in the fact that economic belts such as the Yangtze River Delta and the Pearl River Delta have formed significant characteristics of talent agglomeration. The spatial spillover effect of ER is negative (−0.858)

at 5% level. It indicates that intensifying environmental regulation in adjacent regions can decrease the employment scale of the local region. According to the substitution effect and spatial effect, this may be due to the fact that when ER are strengthened in surrounding areas, firms will increase employment demand in order to comply with the regulations, leading to the loss of local talents.

Among the control variables, human capital increases employment scale in the FE, SAR and SEM models. This is not different from the results of Naval et al. [53]. In line with Appel et al. [54], patents positively affect the local employment scale in the three models. When a new technology is developed, there is the tendency that employment would increase due to the fact that labor would be needed to monitor its operation. Nevertheless, the spillover effect of human capital to surrounding areas is not significant. What this suggests is that governments in neighboring areas should actively develop policies to absorb the inflow of talent. The coefficient of $W \times lnGDP$ is significant and negative ($-0.988$). This is an indication that when there is an improvement in economy of the surrounding provinces, there is high possibility that a chunk of the labor force from local province would be attracted to the surrounding provinces. Patent rather exerts a spillover influence on expanding the local employment scale at 1% level. It is possible that the inter-regional technology spillover effect increases the technology level in both local and surrounding regions, which serves as an avenue for new job opportunities creation.

To further explore the impact of ER on employment, we decomposed the effects of the explanatory variables into direct and indirect effects (see Table 4). In the SDM model, the direct and indirect effects of ER on the employment scale are respectively positive and negative, and the total effect is negative. This result proves that the strengthening of local ER will reduce employment in the surrounding area. This may be due to the unique governance model of local environmental tournaments under central supervision. When ER are strengthened in a local province, neighboring regions will also increase the intensity of ER in order not to lower their ranking in the environmental championship, which will reduce employment due to the compliance cost effect. For example, under the joint air pollution prevention and control mechanism in the Beijing–Tianjin–Hebei regions, Beijing has continuously evacuated more than 2000 polluting firms since 2013, most of which have moved to Shanxi and Henan provinces, where ER are more relaxed [19]. This will lead to a decline in employment in the Beijing–Tianjin–Hebei regions. The direct and indirect effects of PAT on employment were both significantly positive, indicating that innovation has spatial effect, which is conducive to continuously improving production efficiency and increasing the labor demand of enterprises.

**Table 4.** Effect decomposition of the SDM model at national level.

|  | Direct Effects | Indirect Effects | Total Effects |
|---|---|---|---|
| ER | 0.153 ** | −1.181 ** | −1.028 * |
|  | (0.056) | (0.583) | (0.604) |
| lnHC | 0.148 | 1.796 | 1.944 |
|  | (0.116) | (1.316) | (1.342) |
| lnPAT | 0.085 *** | 1.258 *** | 1.343 *** |
|  | (0.016) | (0.224) | (0.232) |
| lnGDP | 0.200 *** | −1.349 *** | −1.149 ** |
|  | (0.040) | (0.374) | (0.379) |
| lnFDI | −0.031 *** | 0.316 ** | 0.286 ** |
|  | (0.009) | (0.125) | (0.129) |

Notes: *** $p < 0.01$, ** $p < 0.05$, * $p < 0.1$. The Standard errors are in parentheses.

### 4.3. Robustness Checks

In order to confirm the reliability of the results for the ER–employment scale nexus, we conducted a robustness test using the following three methods, as Wang et al. [55]. The first method was the SYS-GMM method, purposefully to check for endogeneity problems [56]. The next test here was using 0–1 matrix and economic matrix to re-evaluate the nexus

between ER and the employment scale to avoid the estimation bias caused by subjective selection of spatial matrix. Then, to alleviate the measurement differences caused by different statistical caliber, the number of employments in different industries is used to replace the number of urban employees. Table 5 presents all the results on the robust check, and the previous results are all confirmed.

**Table 5.** Estimation result of the robustness test.

|  | SYS-GMM | 0–1 Matrix | Economic Matrix | Substitute Variable |
|---|---|---|---|---|
| L.lnEMP | 0.786 *** | | | |
|  | (0.0133) | | | |
| ER | 0.130 *** | 0.181 *** | 0.216 *** | 0.174 *** |
|  | (0.0263) | (0.0498) | (0.0505) | (0.0550) |
| lnHC | 0.344 *** | 0.0975 | 0.439 *** | −0.396 *** |
|  | (0.0395) | (0.1155) | (0.1119) | (0.1230) |
| lnPAT | 0.0397 *** | 0.0544 *** | 0.0815 *** | −0.0419 *** |
|  | (0.0085) | (0.0144) | (0.0145) | (0.0152) |
| lnGDP | −0.0167 | 0.235 *** | 0.112 *** | 0.0173 |
|  | (0.0145) | (0.0399) | (0.0395) | (0.0432) |
| lnFDI | 0.0205 *** | −0.0302 *** | −0.0188 ** | −0.0325 *** |
|  | (0.0045) | (0.0077) | (0.0079) | (0.0085) |
| $\rho$ | | 0.340 *** | 0.277 *** | −0.364 ** |
|  | | (0.0403) | (0.0542) | (0.1564) |
| W×ER | 6.968 *** | −0.224 ** | −0.274 ** | 2.633 *** |
|  | (1.8769) | (0.0869) | (0.1203) | (0.3881) |
| W×lnHC | 5.268 *** | −0.0146 | −0.27 | 0.429 |
|  | (0.6576) | (0.1982) | (0.3169) | (0.8568) |
| W×lnPAT | 0.6418 | 0.159 *** | 0.184 *** | −0.0844 |
|  | (0.5707) | (0.0267) | (0.0313) | (0.0952) |
| W×lnGDP | −6.2303 *** | −0.228 *** | 0.0375 | 0.439 * |
|  | (1.3728) | (0.0640) | (0.0912) | (0.2269) |
| W×lnFDI | 2.7892 *** | 0.00921 | −0.202 *** | −0.479 *** |
|  | (0.5740) | (0.0147) | (0.0219) | (0.0811) |
| Model | FE | FE | FE | FE |
| Observations | 660 | 690 | 690 | 690 |

Notes: *** $p < 0.01$, ** $p < 0.05$, * $p < 0.1$. The Standard errors are in parentheses.

*4.4. Heterogeneity Analysis*

Considering the huge regional differences, the spatial impacts of ER on China's employment might be inconsistent with the results for national level. Hence, there is a need to carry out heterogeneity analysis and re-estimate their spillover effects. The samples were divided into coastal and inland areas according to the location. We also looked at the degree of pollution, therefore categorizing them into heavy and less pollution areas according to the degree of environmental pollution for heterogeneity analysis (see Table 6).

In coastal areas, there is not much of visible impact on the employment–environmental regulation nexus. However, when environmental regulation is strengthened in the surrounding areas, local employment declines. The influence of HC and PAT on the local and adjacent employment scale is positive at 1% level. The reason is that enriched human capital on the Coast of China is capable of developing new technologies. This invariably creates more jobs for local and surrounding provinces. Increasing foreign direct investment could decrease employment. This is because advanced manufacturing squeezes jobs from the low-skills labor force, leading to a reduction in the size of employment. In the in-land areas, ER has a significant positive impact on local labor demand, while the same has no spatial effect on employment in the neighboring areas. The effect of human capital and patent application remains consistent and still significantly improves the employment ratio in the local and surrounding areas.

**Table 6.** Spatial analysis at regional level.

| | Coastland | Hinterland | Heavy Pollution | Less Pollution |
|---|---|---|---|---|
| | SDM | SDM | SDM | SDM |
| ER | 0.0472 | 0.144 ** | −0.00028 | 0.427 *** |
| | (0.0832) | (0.0600) | (0.0638) | (0.1216) |
| lnHC | 3.012 *** | 0.245 ** | 0.22 | 0.214 * |
| | (0.3019) | (0.0955) | (0.2404) | (0.1114) |
| lnPAT | 0.0883 *** | 0.0966 *** | 0.140 *** | 0.115 *** |
| | (0.0282) | (0.0125) | (0.0190) | (0.0215) |
| lnGDP | 0.272 *** | 0.252 *** | 0.534 *** | −0.137 ** |
| | (0.0940) | (0.0325) | (0.0547) | (0.0565) |
| lnFDI | −0.0632 *** | 0.00984 | −0.122 *** | 0.011 |
| | (0.0205) | (0.0068) | (0.0139) | (0.0103) |
| $\rho$ | −0.522 *** | −0.638 *** | −19.19 * | −0.125 |
| | (0.1487) | (0.1970) | (11.7382) | (0.1202) |
| W×ER | −1.069 *** | 0.176 | −15.28 | −58.26 *** |
| | (0.3575) | (0.4090) | (22.3086) | (17.7544) |
| W×lnHC | 10.78 *** | 3.700 *** | −17.42 | −15.21 |
| | (1.1759) | (0.6252) | (33.0861) | (26.5870) |
| W×lnPAT | 0.412 *** | 0.446 *** | 21.66 *** | 37.41 *** |
| | (0.0923) | (0.0910) | (5.1461) | (4.9646) |
| W×lnGDP | −0.908 *** | 0.445 ** | −2.142 | −52.47 *** |
| | (0.3189) | (0.1907) | (10.3847) | (7.8012) |
| W×lnFDI | −0.0283 | 0.280 *** | −19.07 *** | 22.04 *** |
| | (0.0782) | (0.0504) | (4.9482) | (3.3691) |
| Log-likelihood | 212.1713 | 543.0930 | 314.1339 | 355.6009 |
| Model | FE | FE | FE | FE |
| Observations | 253 | 437 | 345 | 345 |

Notes: *** $p < 0.01$, ** $p < 0.05$, * $p < 0.1$. The Standard errors are in parentheses.

From the angle of solid waste and exhaust gas outputs of each province, this research categorizes the sample into heavy pollution region and less pollution region for regression analysis. Although ER has a negative relation with employment in heavy pollution areas and adjacent areas, it is not statistically significant. Strict ER has a significant positive influence on employment in lower-pollution areas and a significant spillover effect on decreasing employment in neighboring areas.

*4.5. Further Analysis*

To evaluate the spatial effects of heterogenous ER on employment scale, we further analyzed the employment impacts of CER, MER and VER (see Table 7). As seen in the column (1), the role of CER on local employment is positive but not significant. When environmental regulation is reinforced in neighboring provinces, it would negatively affect local employment. This is consistent with the conclusion of Zhang et al. [19]. The "one size fits all" of CER has contributed to this and reduced nearby employment due to industrial relocation. As seen in column (2), MER improves employment in both local and adjacent regions, indicating that the solidification of MER in neighboring provinces may lead to "Race to the top", which increases employment in both local and surrounding provinces. This is consistent with Yang et al. [14] and Bu et al. [15]. The innovation compensation hypothesis held that ER is regarded as the driving force for innovation. Innovation increases productivity, and firms hire more workers to expand production. Column (3) reports that the influence of VER on labor demand in local and neighboring provinces and cities is insignificantly positive. It indicates that there is still room for improvement for VER to make the necessary impact in China. As it stands, although its impact is positive, however, it is unable to exert innovative compensation effect. Column (4) shows that the three categories of ER as outlined in this study have a positive impact on local employment; however, MER is the only one with a significant effect. Meanwhile, CER has negative spatial effect, while

MER and VER have positive spatial effect. The policymakers should follow the market laws and guide them, implement appropriate market policies, and coordinate the nexus between economic development and environmental protection.

**Table 7.** Effect of different environmental regulations on employment.

|  | SDM | SDM | SDM | SDM |
|---|---|---|---|---|
| CER | 0.052 | | | 0.0117 |
|  | (0.0417) | | | (0.0395) |
| MER | | 0.588 *** | | 0.586 *** |
|  | | (0.0553) | | (0.0553) |
| VER | | | 0.0634 | 0.0477 |
|  | | | (0.0579) | (0.0542) |
| lnHC | 0.0816 | 0.226 ** | 0.121 | 0.226 ** |
|  | (0.1175) | (0.1101) | (0.1187) | (0.1100) |
| lnPAT | 0.0719 *** | 0.0494 *** | 0.0742 *** | 0.0513 *** |
|  | (0.0147) | (0.0137) | (0.0147) | (0.0138) |
| lnGDP | 0.237 *** | 0.252 *** | 0.218 *** | 0.266 *** |
|  | (0.0417) | (0.0392) | (0.0423) | (0.0390) |
| lnFDI | −0.0354 *** | −0.0330 *** | −0.0371 *** | −0.0394 *** |
|  | (0.0081) | (0.0076) | (0.0086) | (0.0079) |
| ρ | 0.315 *** | 0.204 | 0.332 *** | 0.174 |
|  | (0.1121) | (0.1239) | (0.1101) | (0.1269) |
| W×CER | −1.067 *** | | | −1.012 *** |
|  | (0.2777) | | | (0.2618) |
| W×MER | | 1.562 *** | | 1.623 *** |
|  | | (0.4317) | | (0.4286) |
| W×VER | | | 0.226 | 0.959 *** |
|  | | | (0.3961) | (0.3701) |
| W×lnHC | 0.878 | 1.323 * | 1.602 * | 1.648 ** |
|  | (0.7896) | (0.7281) | (0.8419) | (0.7791) |
| W×lnPAT | 0.835 *** | 0.577 *** | 0.772 *** | 0.617 *** |
|  | (0.0935) | (0.0921) | (0.0928) | (0.0931) |
| W×lnGDP | −1.052 *** | −0.393 * | −0.891 *** | −0.470 ** |
|  | (0.2188) | (0.2183) | (0.2198) | (0.2193) |
| W×lnFDI | 0.248 *** | 0.106 | 0.162 ** | 0.0771 |
|  | (0.0757) | (0.0705) | (0.0814) | (0.0760) |
| Model | FE | FE | FE | FE |
| Observations | 690 | 690 | 690 | 690 |

Notes: *** $p < 0.01$, ** $p < 0.05$, * $p < 0.1$. The Standard errors are in parentheses.

Table 8 reports the results of heterogeneous ER on employment in different regions. The coastal region's MER can significantly increase local employment, while CER exerts a significant negative influence on job creation in neighboring areas. This may be due to the strong innovation ability of enterprises in coastal areas. Strict environmental regulation stimulates the innovation compensation effect in coastal areas, thus attracting the labor force in surrounding areas. Different from the results in the coastal region, the CER exerts a significant and positive influence on job recruitment in the in-land region. The direct and spatial effect of MER is significant positive on the employment scale.

On the pollution intensity premises, the heavy pollution region's MER exerts a positive influence on employment scale, while CER exerts negative influence on employment scale. This suggests that CER issued by the government would affect negatively local employment, but MER, on the other hand, would increase employment and better sustain economic growth. Interestingly, the spatial effects of the three categories of ER per this study are not significant. This indicates that three regulation tools in heavily polluted areas are independent, and there is no spatial spillover effect.

**Table 8.** Effect of different environmental regulations on employment for sub-sample.

|  | Coastland | Hinterland | Heavy Pollution | Less Pollution |
|---|---|---|---|---|
| CER | −0.0113 | 0.181 *** | −0.135 *** | 0.103 |
|  | (0.0608) | (0.0463) | (0.0420) | (0.0861) |
| MER | 0.508 *** | 0.233 *** | 0.601 *** | 0.417 *** |
|  | (0.0917) | (0.0847) | (0.0607) | (0.0990) |
| VER | −0.114 | −0.0933 | 0.0704 | 0.416 *** |
|  | (0.0947) | (0.0603) | (0.0691) | (0.1206) |
| lnHC | 2.939 *** | 0.236 ** | 0.157 | 0.186 * |
|  | (0.2870) | (0.0925) | (0.2100) | (0.1084) |
| lnPAT | 0.0518 * | 0.0845 *** | 0.123 *** | 0.0966 *** |
|  | (0.0282) | (0.0130) | (0.0178) | (0.0211) |
| lnGDP | 0.350 *** | 0.286 *** | 0.516 *** | −0.106 * |
|  | (0.0903) | (0.0322) | (0.0480) | (0.0569) |
| lnFDI | −0.0394 * | 0.00634 | −0.0931 *** | 0.00629 |
|  | (0.0219) | (0.0070) | (0.0129) | (0.0101) |
| $\rho$ | −0.561 *** | −0.647 *** | −17.56 | −9.865 |
|  | (0.1493) | (0.2006) | (14.8772) | (8.5308) |
| W×CER | −0.561 ** | −0.0172 | 7.494 | −31.69 ** |
|  | (0.2544) | (0.3102) | (17.4574) | (12.9196) |
| W×MER | 0.4 | 0.979 * | 9.411 | −61.02 *** |
|  | (0.3925) | (0.5374) | (18.6350) | (22.2945) |
| W×VER | −0.192 | 0.648 | −9.663 | 6.293 |
|  | (0.3274) | (0.4511) | (15.1252) | (25.5061) |
| W×lnHC | 9.832 *** | 3.843 *** | −39.92 | −1.1 |
|  | (1.1236) | (0.6134) | (30.0395) | (27.1997) |
| W×lnPAT | 0.354 *** | 0.368 *** | 9.828 * | 36.93 *** |
|  | (0.0964) | (0.0963) | (5.3545) | (5.2078) |
| W×lnGDP | −0.675 ** | 0.412 ** | 12.45 | −50.97 *** |
|  | (0.3106) | (0.1913) | (9.5960) | (7.8925) |
| W×lnFDI | 0.0343 | 0.239 *** | −17.07 *** | 20.45 *** |
|  | (0.0893) | (0.0522) | (4.6659) | (3.4189) |
| Model | FE | FE | FE | FE |
| Observations | 690 | 690 | 690 | 690 |

Notes: *** $p < 0.01$, ** $p < 0.05$, * $p < 0.1$. The Standard errors are in parentheses.

The lower-pollution region, on the other hand, has MER and VER exhibiting positive influence on employment scale; both CER and MER have negative spatial effects on employment. This may occur as the positive effect of job growth from technological innovation resulting from environmental regulation. The same ER offset job losses in high polluting industries. Patents play a positive role in increasing China's employment in all regions.

## 5. Conclusions and Policy Implications

Based on the SDM model, this study examines the impact of China's ER on employment and draws some interesting conclusions. First, ER has a positive impact on local employment and a negative impact on the surrounding area. This is probably due to a combination of substitution effects and innovation effects. In addition, as the intensity of local ER is strengthened, there will be positive spillover effects on surrounding ER under the Regional Joint Prevention and Control and environmental tournament evaluation mechanism. Second, the heterogeneity analysis shows that the spatial effect of ER on employment is significantly negative in the coastal area and lower-pollution regions. It is also found that there is a significant direct impact of ER on expanding local employment scale in in-land areas and lower-pollution regions. Interestingly, MER can significantly increase the employment scale in local and neighboring areas, while the spatial impact of CER is significantly negative. Under the output effect, CER may force firms to reduce scale, close or relocate. While MER can help firms improve competitiveness and expand market share, firms have more incentive to expand scale, increasing demand for labor.

The conclusions of this paper have important policy implications for building an environmental regulation system and expanding employment effects. First, ER have spillover effects on employment in surrounding areas. The local governments should fully consider the synergistic effect of employment in the surrounding areas when formulating ER. In order to avoid the reduction of employment caused by the pollution's nearby relocation, the same intensity of ER can be applied as in the surrounding areas. Meanwhile, in the implementation of environmental governance, it is necessary to implement the safeguard measures of steady employment. For example, improve the green finance system, encourage green transformation of enterprises, and create new employment demand through green innovation and industrial upgrading. In addition, the government should increase the re-education and training of workers and promote re-employment by improving workers' skills.

Second, for in-land or less-polluted areas, the government should develop eco-agriculture or eco-tourism and reduce the undertaking of pollution transfer enterprises. The government should set environmental technology thresholds when attracting investment to ensure that foreign investment or industrial transfer is conducive to the sustainable development of the local economy and avoid the old path of "pollution first, treatment later". For highly polluted areas and coastal areas, strict ER should be implemented to force polluting enterprises to transform into green industries and increase the demand for highly skilled-labor.

Third, policymakers should coordinate different ER tools according to regional characteristics, rather than taking a "one size fits all" approach. For regions with a high degree of marketization, the carbon emission trading mechanism should be improved to give full play to the innovation effect and employment effect of MER. In addition, in areas with good environmental governance effects, channels for public environmental participation should be unblocked to provide policy support for VER to play its role and form a diversified ER system. Our findings suggest that CER have a negative effect on employment, and that policymakers need to consider these adverse factors to achieve the UN sustainable development goals on employment and the environment.

This study also has some limitations. For example, we only looked at the effect of ER on employment scale, and it would have been interesting to explore more heterogeneous labor effects. However, due to the availability of data, this paper was unable to explore the influence of ER on labor with different skills, which will be a potential research topic in future. In addition, it is also necessary to explore interaction between ER and innovation to reveal the impact mechanism of ER on employment. Finally, provincial panel data were used in this study. More detailed data of cities, districts and counties could be used in further research, which would have more important reference value for decision makers.

**Author Contributions:** C.W.: Conceptualization, Methodology, Funding acquisition, Supervision, Reviewing and Editing. Y.H.: Methodology, Software, Data curation, Writing—Original draft preparation, Visualization, and Investigation. All authors have read and agreed to the published version of the manuscript.

**Funding:** This study is supported by National Natural Science Foundation of China (No. 71803068), China Postdoctoral Science Foundation (No. 2019M651387) and the Graduate Research and Innovation Project of Jiangsu Province Funding (No. KYCX21_3300).

**Institutional Review Board Statement:** No animal or human parts were used in this study. The manuscript has not been published elsewhere, and it has not been submitted simultaneously for publication elsewhere.

**Informed Consent Statement:** Not applicable.

**Data Availability Statement:** Data used in this paper are available in China Statistical Yearbook, China Environmental Statistical Yearbook.

**Conflicts of Interest:** The authors declare no conflict of interest.

## Appendix A

**Table A1.** Evaluation index system for China's environmental regulation.

| Primary Indices | Secondary-Class Indices | Third-Class Indices |
| --- | --- | --- |
| Environmental regulation | CER | the number of projects implementing the environmental impact assessment system |
| | | the number of environmental protection administrative punishment cases |
| | MER | the ratio of sewage fee in industrial added value |
| | | the ratio of industrial pollution control investment in industrial added value |
| | VER | the number of environmental letters and visits |
| | | the number of environmental proposals for National People's Congress and the Chinese People's Political Consultative Conference |

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
