# Peer review of "Does Environmental Regulation Have an Employment Dividend? Evidence from China"

_sustainability, doi:10.3390/su15076307_

Round 1

Reviewer 1 Report

Thank you for the opportunity to review this manuscript. As a researcher in this field myself, I very much enjoyed reading it. This paper examines the spatial spillover effects of environmental regulation on employment scale. While the article is informative and of potential interest among some readers, I found it difficult to justify the rationale and motivation for conducting the research. Below I describe my concerns in greater detail. I hope that my comments are viewed as constructive feedback that will help you clarify and enhance your research quality. comments are as follows:

Firstly, I recommend that you highlight the research contribution/novelty of your work at the end of the abstract to provide readers with a clear understanding of the significance of your study. In addition, I found that the motivation for conducting the research was not sufficiently clear in the introduction, which could be improved by providing more context and background information.

Furthermore, I suggest that you update the literature used in your study and adhere to the referencing style required by the journal. Additionally, the theoretical reasoning of the association examined in your study could be more thoroughly discussed to provide readers with a deeper understanding of the research.

In terms of the results and discussion, I recommend that you rewrite them in a more argumentative manner and use references to support your findings. The current descriptive approach could be improved to enhance the quality and rigor of your research.

It would also be beneficial to discuss the theoretical implications of your study in the conclusion, as well as engaging professional proofreading to ensure that the manuscript is error-free.

I believe your paper has merit and I am certain that it can be significantly improved. Once these comments addressed the paper will make an important contribution to the literature on this subject, Overall, I am delighted to read this manuscript. I hope that my comments and questions will provide the authors with some guidance to improve their manuscript.

Best of luck to you!

Reviewer 2 Report

I believe that the article fulfills the requirements to be accepted, either for its good general presentation (clarity, structure and organization), scientific relevance, theoretical and methodological contribution to the area of knowledge. Furthermore, the subject of the article is very current, important and pertinent, for academics, companies, governments and politics, and consistent with the purpose of the magazine.

The list of bibliographical references moderately represents the state of the art in the subject matter and includes articles published in recent years.

The title reflects what he intended with the article, given the current topic investigated. The introduction meets the minimum criteria of scientific research and the journal and the arguments presented highlight the reasons for the relevance of the study. The theoretical foundation presented seems to me to be sufficient, so there is room for improvement in the field of critical argumentation of the subject under study.

The methodology seems to me to be described in a comprehensive and adequate way to the research problem. However, it seems to me that the model followed could be more described, as well as the authors' reference contemplated in the "References" section.

The discussion is clear and seems to me to be well-founded and the arguments convincing for the progress achieved in the area of knowledge studied. The results have a direct connection with what was exposed in the introduction and literature review.

The conclusion, despite showing that the results have future validity, should be more exploratory of the results of the entire investigation. Because the "Abstract" contemplates (in exaggeration, in my opinion) aspects that should be in the conclusion.

The work is convincing in justifying the filling of the identified gap. The study makes explicit the limitations and presents considerations for future studies.

So, I suggest:

- reformulation of the abstract;

- that the theory of compensation for innovation and the theory of the cost of compliance, the two distinct currents of academic debate, as they refer, were contemplated with emphasis, with due argumentation, in the section of literature review and discussion of the results;

- in the "Introduction" they raise research questions, which seems to me not the most appropriate. Shouldn't these be raised in the "Literature Review" section? Thus, the "Introduction" section should include, in particular, a brief background to the subject under study, the objective, methodology, main conclusions and structure of the work;

- in the methodology section, a brief framework for "global Moran's I and Geary's C indexes" seems relevant to me (put in the references);

- line 242 - "Table2", the space is missing;

- line 248 - "Table3", the space is missing;

- line 375 - "Table4", the space is missing;

- line 254 - "Fig. 3,", put the word in full;

- line 259 - Where's the figure? To check;

- as mentioned, that the conclusion should be improved, presenting all the conclusions of the study;

- check the authors' referencing rules in the text and in the references section.

Round 2

Reviewer 1 Report

Thank you for giving me the opportunity to review this manuscript again. I would congratulate you on the incredible improvements you have made to your manuscript. The depth and clarity of your research and analysis are truly impressive, and I am confident that they will have a significant impact on your field.

However, I would like to encourage you to take the discussion of the implications of your work even further. While you have touched on some of the potential applications and implications of your findings, I believe there is still much more to be explored in terms of how your research can be applied to real-world problems.

I particularly appreciate the authors' discussion on the research implications of their findings, which provides valuable insights for policymakers to consider the environmental and employment effects of environmental regulation more comprehensively. However, I would like to suggest that the authors further elaborate on these implications and provide specific recommendations for policymakers.

Best of Luck!

Author Response

Dear Reviewers.

Thank you very much for your professional guidance and suggestions, which helped this article a lot. Please see the attachment.

Best wishes

Chao Wu
